# The Structure and Chemical Composition of the Cr and Fe Pyrolytic Coatings on the MWCNTs’ Surface According to NEXAFS and XPS Spectroscopy

**DOI:** 10.3390/nano10020374

**Published:** 2020-02-21

**Authors:** Danil Sivkov, Olga Petrova, Alena Mingaleva, Anatoly Ob’edkov, Boris Kaverin, Sergey Gusev, Ilya Vilkov, Sergey Isaenko, Dmitriy Bogachuk, Roman Skandakov, Viktor Sivkov, Sergey Nekipelov

**Affiliations:** 1Federal State Budgetary Educational Institution of Higher Education “Saint-Petersburg State University”, 199034 St. Petersburg, Russia; a.mingaleva@spbu.ru; 2Komi Science Center Ural Division of the Russian Academy of Sciences, 167982 Syktyvkar, Russia; teiou@mail.ru (O.P.); isaenko@geo.komisc.ru (S.I.); bogachuk108@gmail.com (D.B.); scanick@yandex.ru (R.S.); sivkovvn@mail.ru (V.S.); nekipelovsv@mail.ru (S.N.); 3G.A.Razuvaev Institute of Organometallic Chemistry of the Russian Academy of Sciences, 603950 Nizhny Novgorod, Russia; amo@iomc.ras.ru (A.O.); kaverin@iomc.ras.ru (B.K.); mr.vilkof@yandex.ru (I.V.); 4Institute for Physics of Microstructures of the Russian Academy of Sciences, 603950 Nizhny Novgorod, Russia; gusev@ipmras.ru

**Keywords:** NEXAFS, XPS, MOCVD, total electron yield, multi-walled carbon nanotubes, nanocomposite, (pyrolytic Cr)/MWCNTs, (pyrolytic Fe)/MWCNTs

## Abstract

The paper is devoted to the structure and properties of the composite material based on multi-walled carbon nanotubes (MWCNTs) covered with pyrolytic iron and chromium. Fe/MWCNTs and Cr/MWCNTs nanocomposites have been prepared by the metal organic chemical vapor deposition (MOCVD) growth technique using iron pentacarbonyl and bis(arene)chromium compounds, respectively. Composites structures and morphologies preliminary study were performed using X-ray diffraction, scanning and transmission electron microscopy and Raman scattering. The atomic and chemical composition of the MWCNTs’ surface, Fe-coating and Cr-coating and interface—(MWCNTs surface)/(metal coating) were studied by total electron yield method in the region of near-edge X-ray absorption fine structure (NEXAFS) C1s, Fe2p and Cr2p absorption edges using synchrotron radiation of the Russian-German dipole beamline (RGBL) at BESSY-II and the X-ray photoelectron spectroscopy (XPS) method using the ESCALAB 250 Xi spectrometer and charge compensation system. The absorption cross sections in the NEXAFS C1s edge of the nanocomposites and MWCNTs were measured using the developed approach of suppressing and estimating the contributions of the non-monochromatic background and multiple reflection orders radiation from the diffraction grating. The efficiency of the method was demonstrated by the example of the Cr/MWCNT nanocomposite, since its Cr2p NEXAFS spectra contain additional C1s NEXAFS in the second diffraction order. The study has shown that the MWCNTs’ top layers in composite have no significant destruction; the MWCNTs’ metal coatings are continuous and consist of Fe_3_O_4_ and Cr_2_O_3_. It is shown that the interface between the MWCNTs and pyrolytic Fe and Cr coatings has a multilayer structure: a layer in which carbon atoms along with epoxy –C–O–C– bonds form bonds with oxygen and metal atoms from the coating layer is formed on the outer surface of the MWCNT, a monolayer of metal carbide above it and an oxide layer on top. The iron oxide and chromium oxide adhesion is provided by single, double and epoxy chemical binding formation between carbon atoms of the MWCNT top layer and the oxygen atoms of the coating, as well as the formation of bonds with metal atoms.

## 1. Introduction

Hybrid materials based on multiwalled carbon nanotubes (MWCNTs) with the surface decorated by various metal-containing nanoparticles form a new class of functional nanomaterials. The role of MWCNTs is in their ability to stabilize highly dispersed nanoparticles of metals, metal carbides, or metal oxides, what eventually leads to higher specific surface areas of the deposited nanoparticles. The MWCNTs surface decoration by nanosized metal-containing particles and coatings makes it possible to substantially change their physicochemical properties [1]. These properties, in combination with a large specific surface area, make MWCNTs promising for use as fillers in various polymeric, metal containing, and ceramic matrices. Previously, it was shown that carbon nanotubes exhibit properties similar to those of graphite, in particular, their surface is poorly wetted by metals [2]. This means that to use MWCNTs as reinforcing elements in metal matrix composites, it is necessary to solve the problem of low interfacial adhesion strength. Therefore, the MWCNTs surface modification through inorganic and organic coating deposition is a current problem in modern material science. The solution of this problem is the deposition of nanosized continuous coatings of metal compounds on the outer surface of MWCNT by pyrolysis, having good adhesion to various fillers. In particular, in our previous study the positive effects of coating MWCNTs with titanium carbide nanoparticles on the evolution of the reinforcement structure in bulk aluminum matrix nanocomposites was described [3].

Due to the large number of heterostructures and MWCNTs-based hybrid material synthesis methods (electrochemical methods, physical and chemical vapor deposition techniques, pulsed laser deposition, atomic layer deposition, and others), such materials are the subject of intensive investigations [4]. Great interest in these materials has been stimulated by a wide variety of their applications in photocatalysis, heterogeneous catalysis, gas sensors, and electrochemical capacitors. The modification of the MWCNTs’ outer surface with various metal-containing nanoparticles [5,6] expands the nanotubes functional properties range [7], giving them the necessary magnetic [8], catalytic [9] and electronic [10] properties. A hybrid material based on MWCNTs with a surface decorated by copper nanoparticles (Cu/MWCNTs nanocomposite), was successfully used as a catalyst for the reduction of germanium tetrachloride by hydrogen [11]. It was shown that the application of such a catalyst makes it possible to decrease the process temperature from 1073 K to 723 K and to increase the conversion degree of germanium tetrachloride to 98%. Currently, intensive investigations have been performed on heterostructures prepared by decorating the MWCNTs’ surface with iron nanoparticles in order to obtain a material with the controlled magnetic properties, which can find various applications, for example, for the fabrication of nanoelectronic devices, in magnetic resonance imaging, and magnetic data storage [12]. Composite based on MWCNTs decorated with Fe_3_O_4_ and polyethyleneimine-silver (nanocomposite Fe_3_O_4_-MWCNTs@PEI-Ag) was used as an efficient catalyst for chemoselective reduction of nitroaromatic and nitrile compounds [13]. Similar structures with Fe_3_O_4_ and silver nanoparticles on the of MWCNTs surface were used during the removal of a toxic model dye, methylene blue (MB), and an aromatic nitro compound, 4-nitrophenol (4-NP) [14]. Of great interest are pyrolytic Cr coatings, since they are characterized by high microhardness, heat resistance, hydrophobicity, and chemical resistance to hydrochloric and sulfuric acids and to alkali melt. Catalysts based on chromium oxide have found applications in ethylene polymerization [15], hydrocarbons dehydrogenation [16], methanol selective oxidation [17] and as sensors for ethanol vapor [18].

The development of synthesis methods and the formation regularities investigation of the nanostructured materials based on MWCNTs and inorganic compounds modified by metal-containing coatings and surface nanoparticles (metallic, carbide, or oxide) are relevant and important problems. Regarding MWCNTs, such modification can be implemented by the deposition of continuous 1–50 nm thick metal containing coatings onto the MWCNTs’ outer surface by the metal organic chemical vapor deposition (MOCVD) method. In our previous paper [19] was reported about the possibility of pyrolytic tungsten deposition on the MWCNT surface by the MOCVD method with tungsten hexacarbonyl W(CO)_6_ used as a precursor. It was shown that the tungsten pyrolytic coating on the MWCNT surface is the finely dispersed phase of tungsten carbide WC_(1−*x*)_.

The preparation of a hybrid material based on MWCNTs coated with pyrolytic Cr and Fe thin layers is an actual task in view of such hybrids’ widespread possible application. Despite the intensive investigation of Fe/MWCNT and Cr/MWCNT composites [19,20,21], to date there are many open questions associated with both the heterocomposites synthesis technology and their physicochemical properties diagnostics. In particular, it is important to obtain detailed information about the characteristics of the composite surface and the interaction between the MWCNT outer surface and the metal coating nanolayer. At this stage, the development of informative non-destructive methods for testing and controlling synthesized MWCNTs-based heterostructures with the desired physical properties is an important problem. In order to solve this problem, it is necessary to use a set of complementary research methods. In this work, we performed complex investigations of the synthesized samples by the methods of scanning electron microscopy (SEM), transmission electron microscopy (TEM), X-ray diffractometry (XRD), Raman scattering, ultra-soft X-ray absorption spectroscopy (near-edge X-ray absorption fine structure, NEXAFS) and X-ray photoelectron spectroscopy (XPS) using a Thermo Scientific ESCALAB 250Xi X-ray spectrometer.

Special attention was paid to the investigation of NEXAFS C1s, Cr2p and Fe2p ionization thresholds, because the X-ray diffraction methods for composites structure studying do not allow characterizing the atomic and chemical compositions of the MWCNT outer surface nanolayer’s covering and the MWCNT—(metal coating) interface. It should be noted that NEXAFS spectroscopy methods are promising for the study of these materials, because they make it possible to investigate changes in the composition of subsurface layers with variations in their depth at the nanometer scale, to obtain information about the nearest environment of an atom with the absorbed X-ray quantum, and to test changes in the atomic and chemical compositions of the MWCNT surface and coatings without destruction and modification [21,22,23]. In addition to these capabilities of ultrasoft X-ray spectroscopy, NEXAFS investigations make it possible to determine the effective thickness and chemical composition of the coating layer on the MWCNT surface and to examine their interaction. The studies of the Fe/MWCNTs and Cr/MWCNT composites by NEXAFS and XPS spectroscopy showed that a multilayer structure is formed on the outer surface of the MWCNTs, consisting of the outer surface layer of the MWCNTs, a thin layer of metal carbide coated with an oxide layer. The effective thicknesses of the coating layers are estimated.

## 2. Materials and Methods

### 2.1. Materials

MWCNTs were synthesized in laboratory Hybride Nanomaterials G.A. Razuvaev Institute of Organometallic Chemistry, Russian Academy of Sciences via catalytic chemical vapor deposition (c-CVD) in a tubular quartz reactor. General characteristics of MWCNTs used in the work: average outer diameter 80 nm, average length 300 μm, carbon purity 98 wt.%, Fe-based catalyst residue (5.5–7.5) wt.%.

Toluene is especially pure (JSC ECOS-1, Moscow, Russia), ferrocene ((C_10_H_10_) Fe-98%, Sigma-Aldrich, St. Louis, MO, USA). As an initial chromium organometallic compound to deposit pyrolytic chromium coatings on the MWCNTs surface, a “Barkhos” chromium-organic liquid (COL) was used. COL is a mixture of bis(arene)chromium compounds (the average molecular weight is 292, the total chromium content is 15–16 wt.%), produced by the Kaprolaktam enterprise, Dzerzhinsk, Nizhny Novgorod region (city, nation). Gaseous argon (RF produced by Limited liability company Scientific Production Enterprise “Salyut-gas” TU 2114-011-106-818-63-2005. The argon volume fraction was not less than 99.994%). Iron pentacarbonyl Fe(CO)_5_, Russia, TU 2436-051-05807977-99. 

### 2.2. Experimental Details

#### 2.2.1. Synthesis of Multi-Walled Carbon Nanotubes (MWCNTs)

MWCNTs were synthesized by the MOCVD method with pyrolysis of ferrocene and toluene mixtures in an argon flow (flow rate 500 cm^3^/min) at atmospheric pressure in a tubular quartz reactor in accordance with the procedure outlined in [24]. The MWCNT synthesis method and labware scheme are presented in Appendix A. Figure 1 shows a photograph of MWCNTs in the form of a powder, which was obtained by grinding macrocylinders. Grinding was carried out in a rotary-type disperser (Stegler, Moscow, Russia) with a knife rotation speed of ~400 rpm. The prepared nanocomposite of the MWCNTs coated with a layer of pyrolytic chromium (Cr/MWCNT) was stored in a volume filled with high-purity argon.

#### 2.2.2. Synthesis of Cr/MWCNTs Nanocomposites

Pyrolytic chromium coatings were deposited onto the initial MWCNTs surface as follows. A 50 mg MWCNTs weighed portion was placed into a Pyrex-glass cell 10–12 cm^3^ in volume and heated at 200 °C for one hour with continuous pumping of the volume using a vacuum pump. Then the cell was cooled, filled with argon, and a certain COL volume (from 0.1 to 0.4 cm^3^) was poured into it. The Pyrex cell was cooled in liquid nitrogen, argon was pumped out, and resealed using a gas burner; then, it was placed into a muffle furnace, and the temperature was gradually increased from room temperature to 400°C, which is necessary for high-quality deposition of pyrolytic chromium coatings. 

The decomposition of COL proceeded according to the reaction:
MWCNTs + Cr(C_6_H_6_)_2_ → Cr/MWCNTs + 2(C_6_H_6_)(1)


The Pyrex cell was kept at this temperature for two hours, then muffle furnace heating was turned off, the Pyrex cell was cooled to room temperature, unsealed, and MWCNTs samples with deposited pyrolytic chromium coatings were removed. The nanocomposite Cr/MWCNTs was stored in a volume filled with high-purity argon.

#### 2.2.3. Synthesis of Fe/MWCNTs Nanocomposites

For the pyrolytic iron coatings deposition on the surface of MWCNTs, iron pentacarbonyl Fe(CO)_5_ (weighed from 1 to 12 g) was used. The procedure of the MWCNTs coated with a layer of pyrolytic iron (Fe/MWCNTs) nanocomposites synthesis is described in the Appendix A. The deposition of pyrolytic iron on the surface of the MWCNTs was carried out on the installation, the scheme of which is presented in Appendix A.

The pentacarbonyl iron decomposition proceeded according to the reaction:
(2)$$\mathrm{MWCNTs}+\mathrm{Fe}{\left(\mathrm{CO}\right)}_{5}\;\overset{240\;{}^{{}^{\circ}}C}{\to }\;\mathrm{Fe}/\mathrm{MWCNTs}+5\mathrm{CO}$$


The prepared nanocomposite Fe/MWCNTs was stored in a volume filled with high-purity argon.

### 2.3. Characterization

MWCNTs and hybrid materials synthesized in this work were characterized using various physical and chemical analysis techniques. XRD analysis of the MWCNTs samples and hybrid materials was performed at a Cu*K_α_*-radiation on a Bruker D8 Discover X-ray diffractometer (Bruker Corporation, Billerica, MA, USA) in the *θ*-2*θ* symmetrical geometry with a Gobel mirror, an equatorial Soller slit with the angular divergence of 2.5°, and a slit of 1.5 mm on the primary beam. The obtained diffraction patterns were processed using EVA with the PDF-2 (2012) powder diffraction database.

The surface morphology of the synthesized materials was examined by SEM using Supra 50VP (Carl Zeiss AG, Oberkochen, Germany) scanning electron microscope. The structure of the MWCNTs, pyrolytic chromium and pyrolytic iron coatings grown by deposition from COL and Fe(CO)_5_ on the surface of the MWCNTs was studied by TEM with a LIBRA 200 MC Schottky field-emission gun instrument (Carl Zeiss AG, Oberkochen, Germany) operating at 200 kV and the information resolution limit of 0.12 nm. These equipments are the property of the Center “Physics and technology of micro- and nanostructures” at Institute for Physics of Microstructures of Russian academy of sciences.

#### 2.3.1. Raman Spectroscopy

Raman studies were conducted using a Horiba-Yvon Jobin LabRam HR800 spectrometer (Horiba, Ltd., Kyoto, Japan) at room temperature and an Ar laser as laser excitation source with a 1 mW at 488 nm wavelength. Samples were analyzed with 50× and 100× lenses. To limit the power of the laser radiation, neutral filters were used. Spectra were recorded in the 100–4000 cm^−1^ range using a spectrometer grating of 600 g/mm (Horiba Ltd., Kyoto, Japan). The spectral and spatial resolution was about 1 cm^−1^ and 1 μm, respectively. Each spectrum was the result of three accumulations with a 10 s exposure. The spectra were recorded at room temperature. After background correction, individual lines were deconvolved using a curve-fitting procedure from the software provided by LabSpec 5.36.

#### 2.3.2. X-ray Absorption Spectroscopy

The electronic structure of the initial MWCNTs and nanocomposites was characterized by means of near-edge X-ray absorption fine structure (NEXAFS) spectroscopy. All presented data were obtained at the Berliner Elektronenspeicherring für Synchrotronstrahlung (BESSY) using synchrotron radiation (SR) from the Russian-German beamline (RGBL) [25,26]. This dipole beamline is highly suited for spectroscopic investigations in region of the 100–1400 eV. All spectra were acquired in total electron yield (TEY) mode by recording the sample drain current. Energy calibration was undertaken for each spectrum using the energy separation between the first- and second-order light-excited Au 4f_7/2_ photoemission line taken from a clean gold plate which was additionally mounted to the sample holder. Either, the energy calibrations of investigated NEXAFS C1s spectra were performed using the well-resolved π*-resonance at 285.38 eV on the C1s spectrum of highly oriented pyrolytic graphite (HOPG) [27]. The spectral dependence of the photon flux was determined using a clean Au photocathode. Its cleanliness had been checked before by X-ray photoelectron spectroscopy. The photon flux (in arbitrary units) was determined from the TEY curve of the Au plate by division by the well-known atomic X-ray absorption cross section of Au [28] in accordance with the method described previously in [29]. The photon energy resolution was below 0.05 eV. The samples for absorption measurements were prepared ex situ in air by pressing of powders of the initial MWCNTs and nanocomposites into the surface of clean cupper or indium plate.

#### 2.3.3. X-ray Photoelectron Spectroscopy (XPS)

XPS studies were carried out at the resource center “Physical methods of surface investigation” (Saint Petersburg University Research park). XPS analysis was performed on Thermo Fisher Scientific ESCALAB 250Xi X-ray spectrometer (Thermo Fisher Scientific, Waltham, MA, USA). The X-ray tube with Al*K_α_*-radiation (1486.6 eV) was used as a source of ionizing radiation. The survey spectra and high-resolution single core-level spectra were measured at the pass energies of 100 eV and 50 eV, respectively. To neutralize the charge of the sample during the experiments, an electron-ion charge compensation system was used. The studies were carried out under ultrahigh vacuum 10^−^^10^ mbar at room temperature; in the case of using a sample charge compensation system, the partial pressure of argon in the analytical chamber was 2 × 10^−^^7^ mbar. The experimental data were processed using the ESCALAB 250 Xi spectrometer software (Avantage v5.9904, Thermo Fisher Scientific, Waltham, MA, USA).

## 3. Results and Discussion

The initial MWCNTs samples and the Fe/MWCNTs (the MWCNTs coated with a layer of pyrolytic iron) and Cr/MWCNTs (the MWCNTs coated with a layer of pyrolytic chromium) nanocomposites prepared on their bases were initially tested by SEM, TEM, XRD and Raman spectroscopy. Samples of the nanocomposite Fe/MWCNTs and Cr/MWCNTs were also characterized by selected area electron diffraction (SAED). The scheme of all processing stages is given in Appendix A.

### 3.1. Initial MWCNTs Research

Figure 2a,b show typical SEM images of the initial MWCNTs. Figure 2c–e show typical TEM and HRTEM images of the initial MWCNTs. SEM studies revealed the presence of MWCNTs with different diameters (Figure 2a,b). According to TEM data obtained by the MOCVD method (Figure 2c) carbon nanotubes have the average outer diameter 80 nm, and their length ranged from hundreds of micrometers to several millimeters. A high-resolution TEM (HRTEM) image demonstrates that the lateral surface of the MWCNTs was formed by graphene layers with the distances between them of about 0.34 nm (Figure 2d). The defects in the form of residual graphene layers on the outer surface are also shown. The diameter of the internal channel of the MWCNTs is on average 6–10 nm. The HRTEM image shows the presence of partly filled nanotubes with *γ*-Fe nanoparticles (Figure 2e). The Fourier transform of the HRTEM image of the crystalline iron particle is shown in the insert in the upper right corner of Figure 2e.

The X-ray powder diffraction analysis of the initial MWCNTs samples (Figure 2f) shows that the main signatures (diffractogram details) corresponded to the graphite peaks (002), (100), (101), and (004). On the other side, diffraction peaks (111), (200) confirmed the presence of residual iron catalyst (*γ*-Fe). An analysis of the peaks’ half-widths showed that the particle sizes of these crystalline phases do not exceed 100 nm. Both *γ*-Fe and Fe_3_C phases are localized in the internal volume of MWCNTs. On the MWCNTs diffractogram, the shift of the (002) MWCNTs reflection to a region of smaller angles as compared to graphite was found, which corresponds to an increase in the interplanar distance from 0.336 (graphite) to 0.343 nm (MWCNTs). The (100) peak in the MWCNTs spectrum corresponds to a plane perpendicular to the MWCNT axis. The diffractograms of the studied MWCNTs are in good agreement with published data on the position and diffraction peaks relative intensity [30,31,32].

In order to test the structure of nanotubes, the Raman scattering spectrum was studied. The corresponding experimental curves are shown in Figure 2g. In contrast to the HOPG Raman spectrum, in the MWCNTs spectrum besides the main peak *G* (1579.3 cm^−1^), the peak *D* (1356.8 cm^−1^), which is characteristic of non-crystalline materials, appears. The peak intensities ratio *I*_D_/*I*_G_ reflects the degree of sample graphitization; the lower this value, the higher the graphitization and the fewer defects and impurities in the composition of MWCNTs graphene layers. For the MWCNTs samples used in this study, this ratio is 0.45 and corresponds to high-purity multi-walled nanotubes [33].

Figure 3 shows the NEXAFS C1s absorption spectra of MWCNTs and HOPG for an angle of 40° from the normal to the sample surface. Their clear similarity in the number and energy positions of all the basic peaks of the fine structure demonstrates the structural similarity of the HOPG and MWCNTs surface layers, which are a sequence of graphene layers. It should be noted that the HOPG absorption spectrum obtained at angles close to 45° corresponds to the case of averaging over the incidence angles *θ*.

It can be seen from the figure that the absorption cross section for MWCNTs is lower in the region of the main absorption peaks. However, in the MWCNTs spectrum, in contrast to the HOPG spectrum, the observed structure has less contrast, and the absorption bands are slightly broadened. In particular, the intense *π**-resonance broadens by 0.5–0.7 eV at half of the intensity (see Figure 3), and the energy position of the first *σ**-resonance shifts to the short-wavelength region by ≈0.1 eV. The probable causes of these spectra differences are the curvature of the graphene surface of a nanotube, as well as the presence of defects on the surface and at the ends of the MWCNTs. It is important to note that the NEXAFS of MWCNTs C1s-spectrum is isotropic, i.e., it does not change with variation in the angle of incidence of SR on the surface of the sample under study. The energy positions and relative intensities of the fine-structure elements of the MWCNTs C1s-spectrum are in good agreement with the other studies results [34,35,36,37]. The XPS and NEXAFS study of MWCNTs confirmed the absence of iron compounds on the nanotubes’ surface, but showed the presence of a small number of oxygen atoms. This is probably due to the presence of adsorbed oxygen-containing compounds and carbon oxides on the outer surface of MWCNTs, which results in the appearance of a non-contrast structure in the C1s-spectrum NEXAFS in the range 287–290 eV, typical for carbon oxides [34]. This is confirmed by the XPS measurements of MWCNT.

### 3.2. Cr/MWCNTs Nanocomposite Research

Figure 4a–e shows the results of the SEM and TEM studies of the MWCNTs coated with pyrolytic chromium. It was found that the composite morphology strongly depends on the mass of pyrolytic chromium coating and deposition conditions. If the mass ratio of the coating and MWCNT is much less than 1:1, there is a tendency to form a continuous pyrolytic chrome coating with a thickness of about 20 nm (Figure 4a). When the ratio increased to 1:1 and more, the bead-shaped structures formed on the MWCNTs surface were detected (Figure 4b). According to TEM and SAED (selected area electron diffraction) the pyrolytic chromium coating is amorphous. In the SAED image (Figure 4d) there are MWCNTs arc reflexes and an amorphous halo.

The X-ray powder diffraction analysis of the initial MWCNTs and composite material samples consisting of MWCNTs with thin and thick pyrolytic chromium coatings showed that they are X-ray amorphous (Figure 4e). To determine the phase composition of the pyrolytic chromium coating the samples were annealed in air at a temperature of 400 °C. Figure 4e shows the diffraction patterns of the Cr/MWCNTs and Cr/MWCNTs nanocomposites after annealing at 400 °C in air. It follows from Figure 4e that the high-temperature samples processing leads to the oxidation of catalytic iron particles located in the MWCNTs channels and the formation of magnetite nanoparticles with a spinel-type structure (Fe_3_O_4_).

### 3.3. Fe/MWCNTs Nanocomposite Research

Figure 5a–f shows the results of the SEM and TEM studies of the MWCNTs coated with pyrolytic iron. It was found that the composite morphology also depends on the mass of pyrolytic iron coating and deposition conditions. SEM images of hybrid material of Fe/MWCNTs with the mass ratio of the coating and MWCNT over 1:1 are demonstrated in Figure 5a–d. In these synthesis conditions MWCNTs had continuous coating of about 10 nm thick. The coating has a rough surface with a granular texture and consists mainly of α-Fe nanocrystallites, and, to a lesser extent, of magnetite (Fe_3_O_4_), which was proved by the SAED methods (Figure 5f) and XRD (Figure 5g). According to X-ray powder diffraction analysis results, the main crystalline phases of the coating are α-Fe and magnetite with a spinel-type structure (Fe_3_O_4_).

### 3.4. Raman Spectra of the Nanocomposites and Initial MWCNTs

Figure 6 shows the Raman spectra of the nanocomposites and initial MWCNTs. It can be seen from the figure that the nanocomposites spectra contain all the characteristic peaks of the initial nanotube with the intensity ratio I_D_/I_G_ close to 0.5. This suggests that during the deposition of metals, the nanotubes structure does not collapse. At that, in the Fe/MWCNTs Raman spectrum, three weak additional peaks are observed at ~297 cm^−1^, ~550 cm^−1^ and ~670 cm^−1^, which are characteristic of Fe_3_O_4_ oxide [38]. Their appearance indicates the complete oxidation of the thin coating of pyrolytic iron. In this case, it is reasonable to assume the oxidation of pyrolytic chromium in the Cr/MWCNTs nanocomposite and expect the appearance of peaks characteristic of Cr_2_O_3_ in its Raman spectra. However, this is not observed due to the nanoscale thickness of the chromium coating, its amorphous nature, and the small absorption coefficient of chromium oxide for 488 nm radiation wavelength, which is approximately five times smaller than that of Fe_3_O_4_ [39,40]. The figure also shows the Raman shift of the Cr/MWCNTs nanocomposite heated in air at a temperature of 400 °C. It can be seen from the figure that the spectrum of the heated sample contains narrow peaks of ~305 cm^−1^, ~348 cm^−1^ and ~551 cm^−1^, corresponding to the two *E*_g_ modes and intense A_1g_ mode characteristic of the crystalline Cr_2_O_3_ phase [41,42,43]. Along with this structure, the spectrum has a peak of ~677 cm^−1^, the appearance of which, according to XRD data, is associated with the iron oxidation and the Fe_2_O_3_ particles formation during the heating of the iron catalyst particles inside the nanotube.

### 3.5. Near-Edge X-ray Absorption Fine Structure (NEXAFS) Spectroscopy Research

The surface of MWCNTs, their chemical composition, structure, layers thicknesses in a coating, and the interaction between the coating layer and MWCNTs for Fe/MWCNTs and Cr/MWCNTs nanocomposites were studied using ultra-soft X-ray spectroscopy methods (NEXAFS spectroscopy and XPS). The absorption cross section spectral dependences in relative units in the wide energy interval 250–900 eV and in the C1s, Cr2p, and Fe2p absorption edge regions of the initial MWCNTs and nanocomposites were measured in TEY mode. The quantum yield method of an external X-ray photoelectric effect for SR studies was first proposed in [44] and now it is widely used for NEXAFS studies in all synchrotron centers in various modifications, including the TEY mode.

Accurate calculations of the TEY signal are very difficult to perform due to the complexity of describing the scattering processes and avalanche formation of electrons. However, the TEY expression for conductors, in the framework of a simplified model, when the scattering processes are described by the mean free path λ, which does not depend, according to a first approximation, on the energy of secondary electrons, can be written in the form [45,46]:
*Y_s_* ~ *I*_0_*E*_0_*σ_s_*(*E*_0_) *φ_s_* (*E*_0_),(3)
where *I*_0_ is the intensity of the incident X-ray radiation with energy *E*_0_, and *σ_s_*(*E*_0_) is the absorption cross section and *φ_s_*(*E*_0_) is a function that monotonically depends on the energy of the incident quantum and is determined for each specific sample *s*.

The incident radiation contains a non-monochromatic background, which consists of scattered long-wavelength vacuum ultraviolet (VUV) radiation and radiation of multiple orders of reflection from the diffraction grating. Tor successfully use of the TEY method for measuring absorption cross sections, it is necessary to measure and suppress this nonmonochromatic background. This is important when using high-intensity SR, since in this case the non-monochromatic background component is large and leads to the appearance of an additional structure in the spectral dependence of the incident SR intensity. This fact creates serious difficulties in normalizing the studied spectra and makes it impossible to measure the absorption cross section and analyze the intensities of the fine structure elements of the X-ray absorption spectra. Taking into account the contribution of background radiation is especially necessary when studying the Cr/MWCNTs nanocomposite, since the NEXAFS Cr2p absorption edges of the pyrolytic chromium coating layer (576–585 eV) in the second diffraction order are superimposed on the structure in the NEXAFS region of the C1s-absorption edge MWCNT (285–295 eV).

In this work, to measure and suppress the contributions of the background radiation in the incident beam and in the recorded TEY signal, an absorption Ti-film filter with a thickness of 230 nm or 150 nm mounted on an Au grid was used. Figure 7 shows the spectral dependence of the absorption coefficient of titanium metal [26,44,45,46] and the ratio of the intensities of the second and third orders to the first diffraction order at the output of the BESSY-2 RGBL channel with a focusing coefficient *C*_ff_ = 2.25 [26]. It can be seen that the background level of multiple orders of magnitude in the region of the C1s absorption edge is at least 10%, and the Ti filter strongly absorbs VUV and short-wavelength second-order radiation in the energy range 455–900 eV.

Figure 8a shows the dependence of *Y*_Au_ of the clean gold surface with and without a Ti filter. The radiation suppression in the region of 453–900 eV, level of non-monochromatic background in the Ti2p edge region (454–460 eV), and background radiation of the second order in the region of 227–450 eV are well observed. It is also can be seen that in the regions of the C1s edge (280–320 eV), N1s edge (390–430 eV) and O1s edge (530–580 eV) there is a structure in the incident beam associated with carbon- and nitrogen-containing impurites and an oxide layer on the surfaces of reflective elements in the SR monochromatization channel.

Figure 8b shows the sequence of taking into account the contribution of this radiation, which was carried out in two stages. At the first stage, the filtration coefficient *β* of short-wavelength radiation was determined when SR was reflected at sliding angles from the surfaces of the monochromator optical elements, which was equal to the ratio of the absorption surges in the Ti2p edge region in the first (454 eV) and second-order (227 eV) diffraction. At the next stage, the radiation spectral dependence of multiple orders of magnitude was determined by dividing the measured TEY signal in the region of 400–1000 eV by *β*, followed by its transfer to the long-wavelength spectral region after a decrease in the energy scale by a factor of 2. The Figure 8b shows that the dependence *Y*_Au_ (curve 2), after subtracting the background contributions from the TEY signal, becomes monotonic in the regions of 225 eV (Ti2p edge second order) and 265 eV (O1s edge second order) (curve 4), this indicates the correct background radiation accounting.

Figure 8c shows the spectral dependence proportional to the incident radiation intensity *I*_0_*E*_0_*φ*_Au_(*E*_0_) in relative units, defined as the ratio of the received monochromatic TEY signal from the Au surface to the gold atom absorption cross section σ_Au_ [28]. The *I*_0_*E*_0_*φ*_s_(*E*_0_) dependences for monoatomic photocathodes made of metallic copper, doped silicon, and highly oriented pyrolytic graphite (HOPG) were determined in a similar way by dividing the TEY signal by atom absorption cross section of copper, silicon, and carbon atoms, respectively [28].

Figure 9 shows the *I*_0_*E*_0_*φ*_s_(*E*_0_) spectral dependences of these photocathodes and gold normalized to a maximum in the region of 450 eV. The dependences of the relative quantities *I*_0_*E*_0_*φ*_Si_(*E*_0_), *I*_0_*E*_0_*φ*_Cu_(*E*_0_), *I*_0_*E*_0_*φ*_MWCNT_(*E*_0_) and *I*_0_*E*_0_*φ*_Au_(*E*_0_) are proportional and, therefore, the ratio of the functions *φ*_s_(*E*_0_) and *φ*_Au_(*E*_0_) is constant in the spectral range 200–600 eV. This means that the TEY signal from MWCNTs and nanocomposites based on them can be normalized to the incident radiation intensity using the gold TEY signal divided by the absorption cross section of the Au atom to obtain the spectral dependences of the absorption cross sections in relative units.

The effectiveness of the background accounting method was clearly demonstrated by the example of C1s, the spectrum of Cr/MWCNTs and Fe/MWCNTs nanocomposites in Appendix A (Appendix A), which shows the spectra obtained without a titanium filter, with a filter and with subtraction of the multiple diffraction orders background.

Figure 10a shows the absorption cross sections spectral dependences in a single relative scale for a wide spectral region and near the NEXAFS C1s absorption edge for the initial MWCNTs (curve 1) and Fe/MWCNTs (curve 2) and Cr/MWCNTs (curve 3) nanocomposites. These dependences were obtained by dividing the corresponding monochromatic spectral dependences of the TEY signals from composites and MWCNTs by the intensity of the incident SR beam, determined from the *Y*_Au_ signal by the method described above. The dashed lines show the partial absorption cross sections obtained by extrapolating the cross sections from the long wavelength region to the C1s absorption edge. Figure 10b shows the spectral dependences of the C1s partial absorption cross sections of the MWCNTs, the Cr/MWCNTs and Fe/MWCNTs composites. The intensity spectral dependences in arbitrary units near the Fe2p and Cr2p absorption edges of nanocomposites and a number of test compounds are presented in Figure 10c and Figure 10d, respectively. The analysis of the spectral shape and the fine structure elements energy positions of the Fe2p and Cr2p composites spectra clearly demonstrates the formation of Fe_3_O_4_ and Cr_2_O_3_ oxides on the MWCNTs surface. This is evident from a comparison of the NEXAFS spectra of Fe2p oxides Fe_3_O_4_, Fe_2_O_3_, FeO [50] and Cr_2_O_3_, CrO_2_ [51] and CrO_3_ [52]. A similar feature of the nanotube coating by iron atoms is observed, particularly for nanocomposite Fe_3_O_4_-MWCNTs@PEI-Ag [13].

As noted earlier in the analysis of SEM data, the Fe_3_O_4_ and Cr_2_O_3_ coatings are continuous and have good adhesion to the MWCNTs’ outer surface. This suggests a chemical interaction between the oxide layer and the outer graphene layers of MWCNTs. In this regard, is interesting to compare the C1s NEXAFS spectra of the initial MWCNTs and composites. From Figure 10b it can be clearly seen that the structural elements (*π**-and *σ**-resonances) characteristic of the spectrum of the initial MWCNT are retained in the composite spectrum. This indicates that there is no significant destruction of the MWCNTs’ outer layers.

However, in the region of intermediate energies between *π**- and *σ**-resonances (285.4–291.8 eV), there is an additional structure in the form of low-intensity peaks A, B, C, and shoulder D with energies of 287.1 eV, 287.8 eV, 288.4 eV, and 290.4 eV. The energies of these peaks coincide with the energies of the elements in the graphite oxides C1s NEXAFS spectrum [53], and correspond to electron transitions from the C1s-level to *π**-unoccupied orbital of C–O–C, C–O, C=O [54] and [CO_3_]^2−^ [55] atomic groups, respectively. On this basis, the good adhesion of Cr_2_O_3_ and Fe_3_O_4_ to MWCNTs can be explained by the formation of a chemical bond through oxygen atoms between the oxide and the MWCNTs’ outer surface. However, in the initial stage of the 3d-metal atoms deposition on the MWCNTs surface, carbides can form. The C1s spectra of carbides are characterized by the presence of a structure in the energy range 285–277 eV [56].

Due to the mixing of C1s and 3d orbitals (hybridization) of carbon and metal atoms, such a structure also occurs for chromium and iron carbides [57,58]. Due to the superposition of the carbides and metal oxides structures in the NEXAFS C1s X-ray spectra, their study by spectral methods is impossible. However, separate studies of oxides and carbides can be carried out by the XPS method, the results of which will be discussed below. It can be seen from Figure 10b that the integral C1s partial cross sections’ dependences (the area under the curve) of nanocomposites significantly decrease. This is due to the weakening of the flow of photoelectrons and secondary electrons emitted from the MWCNT outer surface in the metal oxides coating layers. These layers are continuous, but not uniform in thickness; therefore, they are characterized by the effective thickness *d*_eff_, which can be determined from the relation:
*d*_eff_ = *λ* Ln(*S*_1_/*S*_2_)(4)
where *λ* is the photoelectron escape depth; *S*_1_ and *S*_2_ are the areas under the C1s partial dependences of the absorption cross sections for the MWCNT and composite, respectively.

Since the NEXAFS signal in the C1s absorption edge region is formed by Auger electrons, whose energies are about 273 eV [59], then according to universal dependence electron mean free paths are 1–2 nm [60]. The S_1_/S_2_ ratio is 2.1 for Cr/MWCNT and 1.21 for Fe/MWCNT, which allows us to estimate the effective thickness minimum of Cr_2_O_3_ and Fe_3_O_4_ coatings; its values are 0.8–1.6 nm and 0.2–0.4 nm, respectively. However, taking into account the contribution of low-energy secondary electrons [45], the effective yield depth can have larger values.

### 3.6. XPS Spectroscopy Research

A more accurate method for determining thickness is XPS. Thickness can be obtained from the change in the XPS C1s peak intensity for MWCNTs compared to nanocomposites. The peak intensity is determined by the yeild of C1s photoelectrons with certain energy equal to the difference between the incident X-ray radiation energy and the C1s electron-binding energy. In the present work, this energy was 1202 eV. Survey photoelectron spectra are given in Appendix A. Figure 11a–c and Appendix A (in Appendix A) show the C1s, O1s, Cr2p, and Fe2p photoelectron spectra of MWCNTs, Cr/MWCNTs, and Fe/MWCNTs. An analysis of the initial MWCNTs spectra shows that the C1s spectrum agrees well with other studies’ data [61,62,63,64,65] at the binding energy of the main peak (284.5 eV) corresponding to the C–C bond and at the wide plasmon band (291 eV), which confirms the high quality of MWCNTs under study. However, in the structure of the O1s spectra of MWCNTs (Figure 11b), two peaks with binding energies of 531.7 eV and 533.5 eV (correspond to the energies of C=O and C–OH bonds, respectively [61,66]) are found. The relative content of oxygen and carbon atoms, determined from a comparison of the integrated peak intensities in the C1s and O1s spectra of MWCNTs, is no more than 3%, which is consistent with Raman and NEXAFS spectroscopy data and demonstrates a very low oxidation state of the nanotubes’ outer surface. Figure 11a,c show the XPS O1s and Cr2p spectra of Cr_2_O_3_ oxide and the Cr/MWCNTs nanocomposite. Both spectra have a common structure in the form of peaks with energies of 575.8 eV and 585.6 eV in the Cr2p spectra and 530.2 eV and 531.6 eV in the O1s spectra. This is consistent with the NEXAFS Cr2p absorption spectrum data and confirms that the coating consists of chromium trioxide Cr_2_O_3_. From the ratio of the C–C bond main peaks’ areas in the spectra of the initial nanotube and Cr/MWCNT nanocomposite *S*_1_/*S*_2_ = 1.9, the effective thickness of the chromium oxide coating layer were determined to be *d*_eff_ = 1.5 nm. In this case, the mean free path of photoelectrons with an energy of 1202 eV in Cr_2_O_3_
*λ* = 2.33 nm [67]. The effective thickness obtained by XPS is in good agreement with NEXAFS measurements.

It should be noted the appearance of two new peaks of 574.6 eV and 584.0 eV in the Cr2p spectra of the Cr/MWCNT nanocomposite, which binding energies are characteristic of 2p_3/2_ and 2p_1/2_ electrons in chromium carbides [67,68]. This definitely indicates the formation of a C–Cr chemical bond and the growth of a chromium carbide layer during pyrolysis. This is confirmed by the appearance of an additional peak of 283.0 eV in the Cr/MWCNT XPS C1s spectrum, the energy position of which agrees with the binding energy of the C1s electron in chromium carbides [68,69]. A clear selection of the peaks related to C–C and C–Cr bonds in the XPS C1s spectrum of the Cr/MWCNT nanocomposite allows the carbide layer thickness in the coating to be estimated as *d*_eff_ = 0.3 nm (the ratio of these peaks areas *S_C_*_–_*_C_*/*S_C_*_–_*_Cr_* equals to 10.4). In this case, it is assumed: (i) the thickness of the chromium carbide layer is small, and the layer is located directly above the MWCNT outer surface; (ii) the cross-section of photoelectron emission from the C1s level is the same for carbon atoms in carbide and MWCNT; (iii) the intensities of the X-ray radiation incident on the surface and on the carbide layer are equal; (iv) the mean free path of an electron in MWCNT at an energy of 1202 eV *λ* = 3.11 nm [67] and is much less than MWCNT thickness. In this case, the effective thickness of the carbide layer can be calculated from the relation [22]:
*d*_eff_ = *λ S*_C–Cr_/*S*_C–C_ = 0.3 nm(5)


In this case the thickness of the oxide layer on the nanotube surface estimated above must be reduced by the found thickness of the carbide layer, since in the first case it was estimated as the entire thickness of the layer covering the MWCNTs.

Thus, the process of forming a chromium atom coating on the MWCNT surface an be divided into three stages: (1) at the initial stage of the MOCVD process, chromium atoms chemically bond with carbon atoms on the surface of the MWCNT, forming a thin layer of chromium carbide 0.3 nm thick; (2) then the chromium carbide layer is covered with a metal layer; (3) upon subsequent removal of the composite into air, the metal layer surface is oxidized to Cr_2_O_3_, the thickness of which is 1.2 nm.

An analogous consideration for the XPS spectra of the Fe/MWCNTs composite suggests a similar mechanism of coating formation on the MWCNT surface (Appendix A in Appendix A). The effective thickness of the Fe_3_O_4_ oxide layer in this case is *d*_eff_ = 0.3 nm, and the effective thickness of the carbide coating is about 0.01 nm.

## 4. Conclusions

It was found that the MOCVD method with pyrolysis of ferrocene and toluene mixtures in an argon stream at atmospheric pressure in a tubular quartz reactor is an efficient and universal way to synthesize the initial MWCNT. SEM, TEM, and XRD studies have shown that the MWCNT outer surface is formed by graphene layers with the distance between them of about 0.34 nm. The MWCNTs’ external diameter and the diameter of their internal channel are 6–10 nm and about 80 nm, respectively. The MWCNTs length varies from hundreds of micrometers to several millimeters. The internal channels contain nanoparticles of crystalline phases *γ*-Fe and Fe_3_C. At the same time, NEXAFS and XPS measurements showed that the MWCNT outer surface remains clean and contains less than 3% of oxygen atoms in the form of carbon oxides. A study of the MWCNTs Raman spectra showed that the peak intensities ratio *I_D_*/*I_G_* is 0.45 and corresponds to high-purity multi-walled nanotubes [33].

It was found that the iron pentacarbonyl [Fe(CO)_5_] and chromium-organic liquid [Cr(C_6_H_6_)_2_] pyrolysis methods for pyrolytic iron and chromium coatings deposition on the surface of MWCNTs make it possible to obtain nanosized metal layers. TEM, SEM, SAED and XRD studies have shown that these layers have a thickness of 10–20 nm, and are continuous and amorphous. Raman and ultra-soft X-ray spectroscopy data showed that after the removal of nanocomposites into the air, metal coatings are oxidized and Fe_3_O_4_ and Cr_2_O_3_ coatings are formed.

The chemical composition analysis, coating thickness measurements, and (pyrolytic metal)/MWCNTs interface study of the nanocomposites, as well as the study of the initial MWCNTs’ surface, were carried out by NEXAFS spectroscopy using SR. A method for taking into account and suppressing non-monochromatic background and multiple reflection orders radiation from the diffraction grating was developed to measure the absorption cross section of X-ray radiation in the region of the NEXAFS C1s absorption edge. In this work, we substantiate this method and demonstrate its effectiveness on the example of Cr/MWCNTs nanocomposite C1s absorption spectra. In this case, taking into account the contribution of background radiation is especially important, since the NEXAFS Cr2p absorption edges of the Cr_2_O_3_ coating layer (576–585 eV) in the second diffraction order are superimposed on the NEXAFS C1s absorption edges of MWCNT (285–295 eV).

NEXAFS and XPS studies have shown that pyrolytic coatings on the MWCNT surface consist of Fe_3_O_4_ and Cr_2_O_3_. It was found that the interface between MWCNTs and the pyrolytic coating (of both Fe and Cr) has a multilayer structure: (i) a layer in which carbon atoms along with epoxy –C–O–C– bonds form bonds with oxygen and metal atoms from the coating layer on the MWCNT outer surface, (ii) a metal carbide monolayer above and (iii) an oxide layer on top. The iron oxide and chromium oxide adhesion is ensured by the formation of a single, double and epoxy chemical bonding between the carbon atoms of the MWCNTs’ upper layer and the oxygen atoms of the coating, and the formation of bonds with metal atoms.

Currently, there are many works on the study of Cr_2_O_3_/MWCNTs composites synthesized by different methods: the sol-gel method [70], metallic chromium evaporation in a helium atmosphere and the chromium clusters condensation on a substrate cooled by liquid nitrogen, followed by the chromium cluster’s oxidation [71], hydrothermal synthesis [72], and synthesis in an autoclave at a temperature of 250 °C in ethanol [73] etc. Chromium oxide samples obtained can have different morphologies, for example, in the form of nanotubes [74], nanocrystalline powder [70,71], porous microspheres [74] and others. Besides, in [18], the chromium-based hybrid material synthesis was carried out by a multistage method in a benzene solution containing chromium acetylacetonate.

In this work, the MOCVD technology for the decomposition of bis(aren)chromium compounds was implemented, which allows the chromium oxide continuous coatings with good adhesion to the MWCNT surface to be obtained. The resulting coating has a multilayer structure. The formation of carbon-oxygen and carbon-metal bonds between the atoms of the coating and the outer graphene layer (with a slight modification of the latter) leads to the formation of a high-adhesion coating.

In addition, a lot of work has been devoted to the synthesis and study of the iron oxides nanoparticles and coatings deposition on the MWCNT surface. In [75], a unique hybrid material CNT/Fe_2_O_3_ was obtained by hydrothermal synthesis method, using polyethylene glycol and FeCl_3_ as precursors at a temperature of 200 °C. In [76], the supercritical solvent technique was applied to obtain a hybrid CNT/Fe_2_O_3_ composite from initial CNTs. Ethanol was used as a solvent, supercritical CO_2_ as an anti-solvent and FeCl_3_ as a precursor. Also for the creation of a hybrid CNT/(pyrolytic iron) iron pentacarbonyl as a precursor can be used [77]. In the work, the technique for the pyrolytic iron coatings deposition during the iron pentacarbonyl thermal decomposition at 205 °C and 30 mTorr pressure is presented. As a result, the CNTs array with an area of 1 cm^2^ was completely covered with 2–3 nm thick pyrolytic iron per one cycle. A simple method has been developed for direct synthesis of magnetic multi-walled carbon nanotubes (MWCNTs) homogeneously decorated with size-controllable Fe nanoparticles encapsulated by graphitic layers on the MWCNT surface by the pyrolysis of ferrocene [12].

In the current work, the MOCVD technology for the iron pentacarbonyl decomposition at a temperature of 240 °C was implemented, which allow us to obtain a continuous Fe_3_O_4_ coating with high adhesion to the MWCNT surface. This coating has a multilayer structure similar to the structure of the chromium coating of Cr/MWCNT nanocomposite. Its high adhesion is also explained by the formation of carbon-oxygen and carbon-metal bonds between the atoms of the coating and the outer graphene layer.

## Figures and Tables

**Figure 1 nanomaterials-10-00374-f001:**
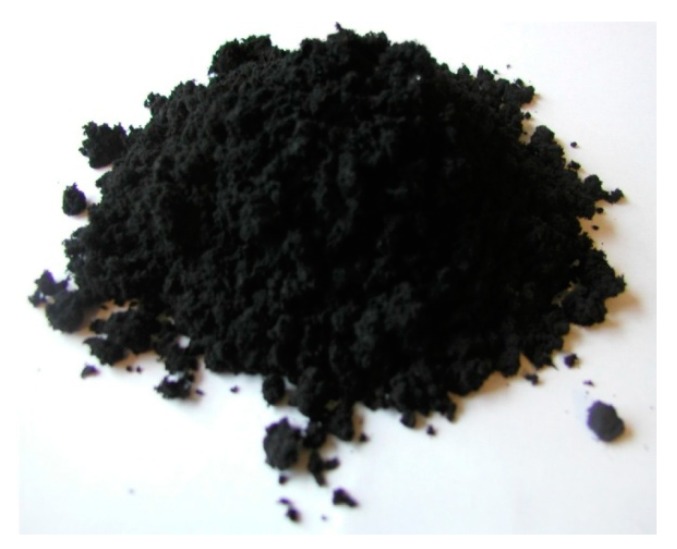
Photograph of multi-walled carbon nanotubes (MWCNTs) powder obtained by mechanical grinding of macro cylinders with walls consisting of radially oriented MWCNTs.

**Figure 2 nanomaterials-10-00374-f002:**
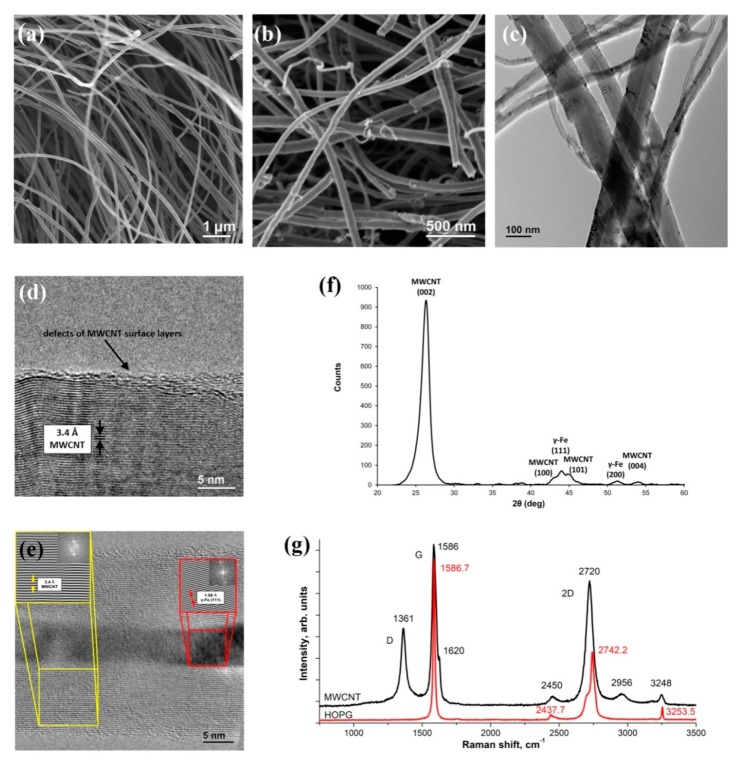
The initial MWCNTs research results: (**a**,**b**) scanning electron microscope (SEM) images of initial MWCNTs; (**c**) transmission electron microscope (TEM) image of MWCNTs; **(d**,**e)** high-resolution TEM (HRTEM) images of a *γ*-Fe-filled nanotube with inserts of FFT (fast Fourier transform) which show the carbon interplanar spacing 0.34 nm and the crystallinity of the filling (*γ*-Fe interplanar spacing 0.198 nm); (**f**) X-ray powder diffraction pattern of initial MWCNTs; (**g**) Raman spectra of MWCNT and HOPG.

**Figure 3 nanomaterials-10-00374-f003:**
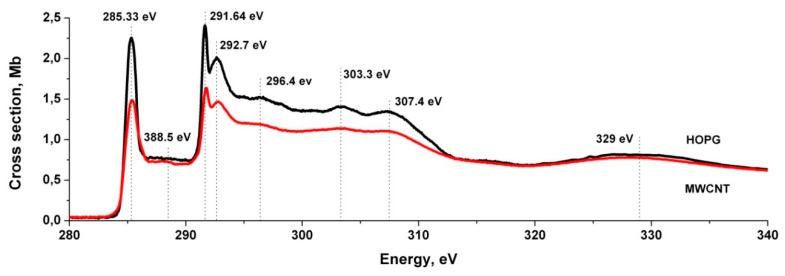
Absorption cross section spectral dependences in the region -edge X-ray absorption fine structure (NEXAFS) C1s edge of MWCNTs (red) and HOPG for an angle of 40° on the normal to the sample surface (black).

**Figure 4 nanomaterials-10-00374-f004:**
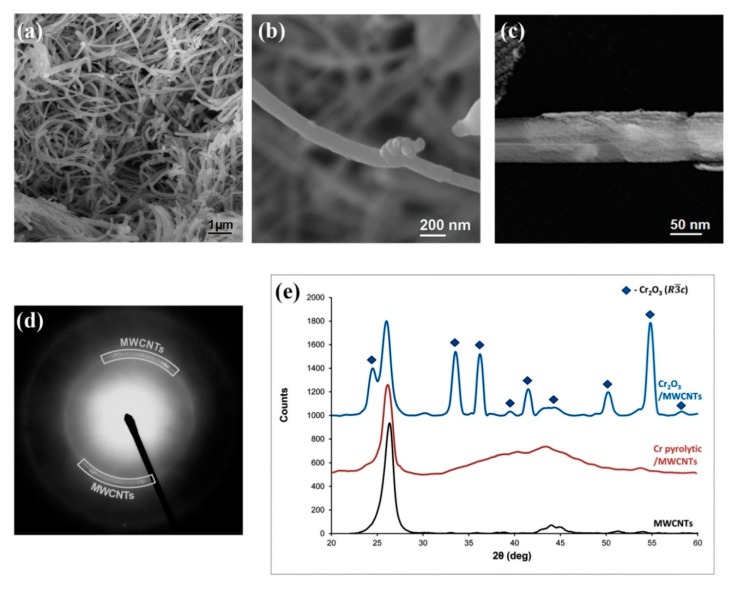
The Cr/MWCNTs nanocomposite research results: (**a**,**b**) SEM and (**c**) TEM micrographs of the Cr/MWCNTs nanocomposite; (**d**) selected area electron diffraction (SAED) image of the Cr/MWCNTs nanocomposite; (**e**) X-ray diffraction (XRD) patterns of the initial MWCNTs (black), the Cr/MWCNTs hybrid material (red), and the Cr/MWCNTs hybrid material after annealing at 400 °C in air (blue).

**Figure 5 nanomaterials-10-00374-f005:**
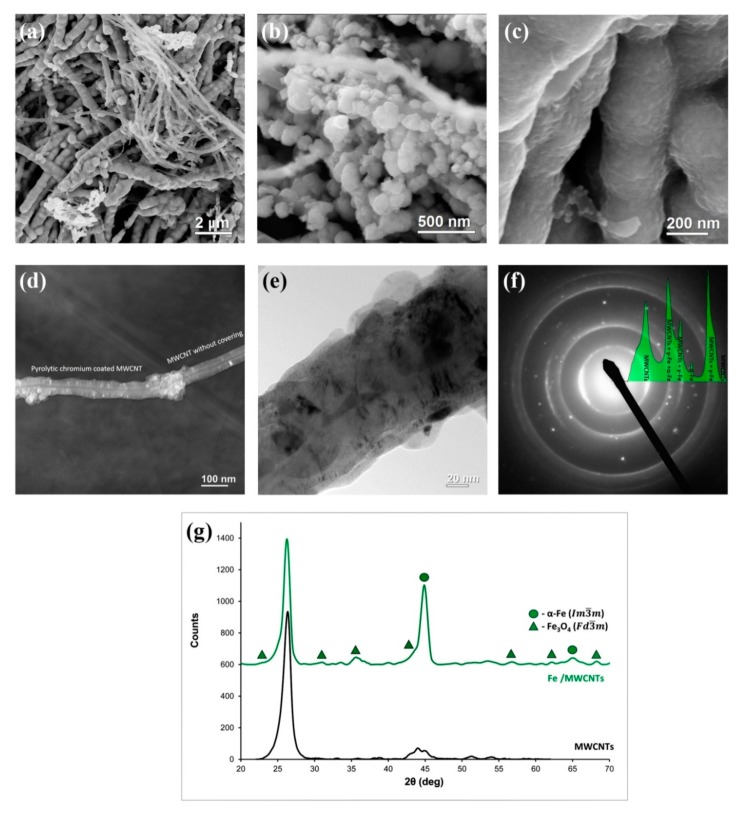
The Fe/MWCNTs nanocomposite research results: (**a**–**c**) SEM micrographs of the Fe/MWCNTs nanocomposite; (**d**,**e**) TEM images of the Fe/MWCNTs nanocomposite; (**f**) SAED with profile of azimuthal integration result; (**g**) XRD patterns of the initial MWCNTs (black) and Fe/MWCNTs nanocomposite (green).

**Figure 6 nanomaterials-10-00374-f006:**
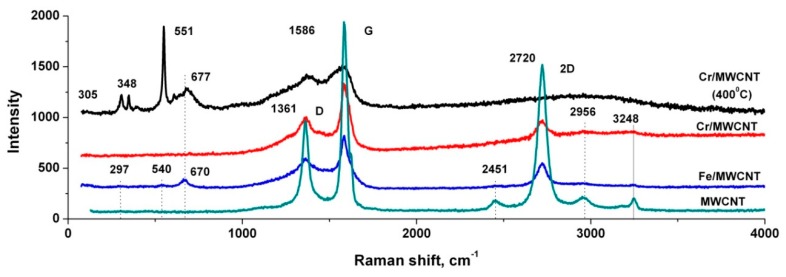
Raman spectra of initial MWCANTs (turquoise), Fe/MWCNTs nanocomposite (blue), Cr/MWCNTs nanocomposite before (red) and after treatment at 400 °C in air (black).

**Figure 7 nanomaterials-10-00374-f007:**
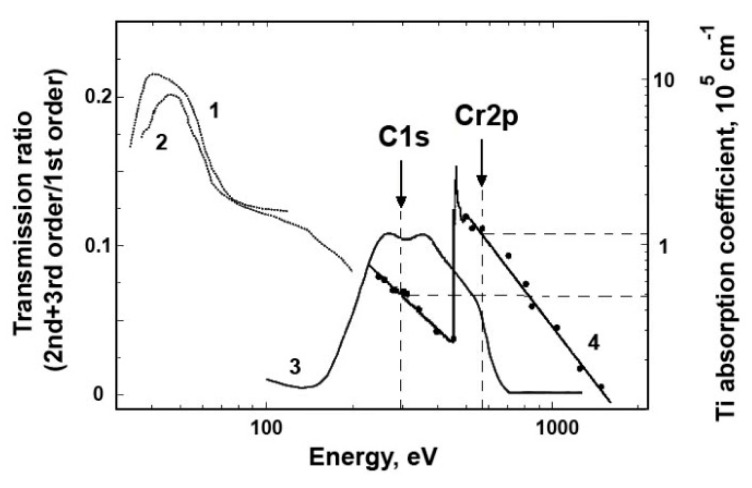
Dependences of the ratio of the higher orders X-ray radiation intensity (2 and 3 orders of magnitude) to the X-ray radiation intensity of the 1st diffraction order for the 1200 lines/mm grating and *C_ff_* = 2.25 [26] (3), and the spectral dependences of the metallic Ti linear absorption coefficient in a wide energy range. Dots and a solid line indicate the results of measurements on X-ray emission lines [29,47] (4), dotted line shows the data in the vacuum ultraviolet area (1) [48], (2) [49]. The arrows indicate the C1s (285 eV) and Cr2p (570 eV) absorption edges positions.

**Figure 8 nanomaterials-10-00374-f008:**
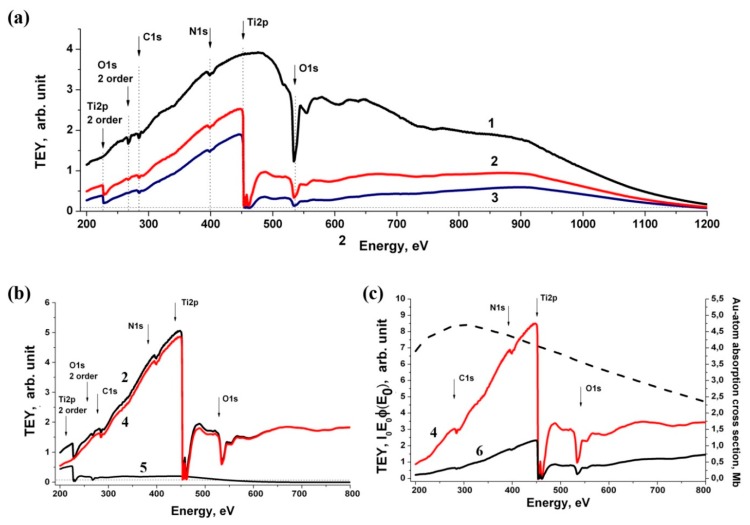
Total electron yield (TEY) signal and *I*_0_*E*_0_*j*(*E*_0_) spectral dependences of pure gold: (**a**) the TEY signal obtained from the surface of pure gold without a filter (1) and using Ti filters with a thickness of 150 nm (2) and 230 nm (3). Vertical lines show the Ti2p, O1s, and N1s absorption edges in the first and second orders of reflection from the diffraction grating; dashed horizontal line is the level of the scattered long-wave background; (**b**) spectral dependences of the measured TEY signal (2), monochromatized TEY signal (4) and second-order radiation (5) on the pure gold surface using 150 nm Ti filter. The arrows indicate the absorption edges in the first and second diffraction orders; (**c**) spectral dependences of the TEY monochromatic signal on the Au plate with a 150 nm Ti filter (4), *I*_0_*E*_0_*j*(*E*_0_) (6) and the absorption cross section of the gold atom [23] (dashed line) in Mb (right scale).

**Figure 9 nanomaterials-10-00374-f009:**
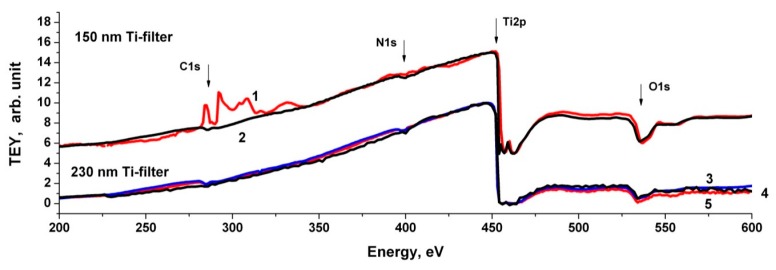
The *I*_0_*E*_0_*j*(*E*_0_) spectral dependences measured by the TEY method using Ti filters with a thickness of 150 nm and 230 nm and various photocathodes: HOPG (1), copper (3), silicon (4) and gold (2.5).

**Figure 10 nanomaterials-10-00374-f010:**
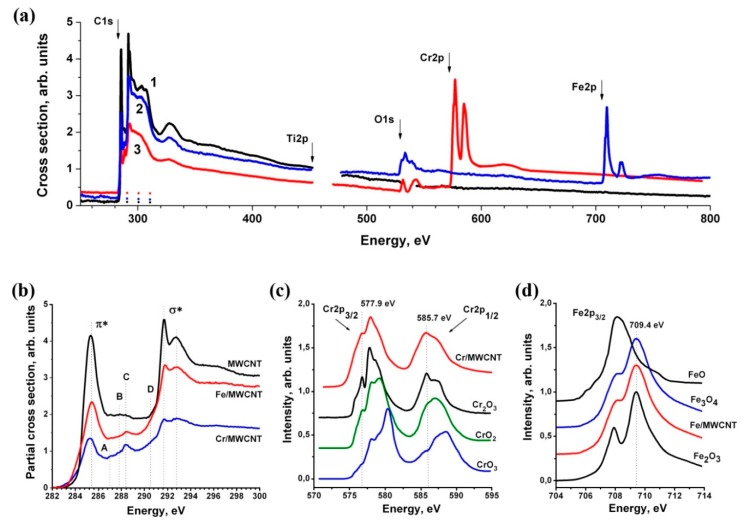
Cross section and intensity spectral dependences in arbytrary units in wide energy range and the region NEXAFS C1s, Fe2p and Cr2p edges: (**a**) The absorption cross section spectral dependences of MWCNT (1), Fe/MWCNT (2) and Cr/MWCNT (3) nanocomposites in relative units. The arrows indicate the 1s absorption edges of carbon, oxygen, and 2p absorption edges of titanium, chromium, and iron; (**b**) the spectral dependences of the partial absorption cross section in the region NEXAFS C1s edge of the initial MWCNTs, Cr/MWCNTs and Fe/MWCNTs nanocomposites in relative units; (**c**) the spectral dependences of the intensity in the region NEXAFS Cr2p edge of the Cr/MWCNTs and chromium oxides Cr_2_O_3_, CrO_3_ [52], and CrO_2_ [51]; (**d**) the spectral dependences of the intensity in the region NEXAFS Fe2p edge of the Fe/MWCNTs composite and iron oxides Fe_3_O_4_, Fe_2_O_3_ and FeO [50].

**Figure 11 nanomaterials-10-00374-f011:**
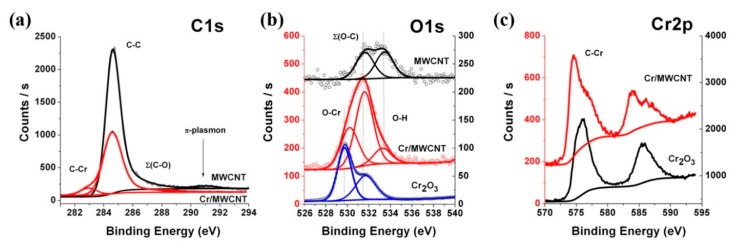
XPS (**a**) C1s, (**b**) O1s and (**c**) Cr2p spectra of the initial MWCNTs, Cr/MWCNTs and chromium trioxide Cr_2_O_3_; (**d**) XPS Cr_2_O_3_ O1s spectrum is reduced by a factor of 10.

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
