# Peer review of "The Structure and Chemical Composition of the Cr and Fe Pyrolytic Coatings on the MWCNTs’ Surface According to NEXAFS and XPS Spectroscopy"

_nanomaterials, 2020, doi:10.3390/nano10020374_

Round 1
Reviewer 1 Report
the paper can be accepted for publication as it is
Author Response
The authors are grateful for the manuscript review.
Reviewer 2 Report
The paper entitled ‘The structure and chemical composition of the Cr and 2 Fe pyrolytic coatings on the MWCNTs surface 3 according to NEXAFS and XPS spectroscopy’ represents very interesting studies covering the complex composite material characterization that includes NEXAFS, XPS, SEM, TEM, RAMAN. The manuscripts require some clarifications and formatting before the recommendation to publish this work, I would like to ask the authors for a few clarifications on the following points:
Adding the schematics of all the processing steps would be useful 4. caption should on page 8, a similar problem with Fig 5. What is the justification to create such a complex composite material? How long does the processing of those materials take? What is the perspective of using these composite?Author Response
Point 1: Adding the schematics of all the processing steps would be useful
Response 1: The processing steps scheme added to the Supplementary (figure S3).
Point 2: 4. caption should on page 8, a similar problem with Fig 5.
Response 2: The drawings were moved one paragraph above. Now the captions is under the figures.
Point 3: What is the justification to create such a complex composite material?
Response 3: The purpose of this work is not only complex material creation, but primarily the development of the methods to study the structures created on the surface of MWCNTs by different methods using the example of Cr/MWCNTs and Fe/MWCNT nanocomposites. In manuscript is shown that the materials under study are complex in structure: a modified upper graphene layer on the MWCNT surface, a metal carbide thin layer above and a oxide coating layer on top. This complex structure expected to be the same in the composite created by other possible methods.
Point 4: How long does the processing of those materials take?
Response 4: The MWCNTs processing is described in detail in [24], what is indicated in the text of the manuscript (page 4, line 148). In addition, the MWCNTs, Fe/MWCNTs and Cr/MWCNTs synthesis processes are described in detail in the Supplementary. Information on the time spent at each stage of the materials syntesis are also available in Supplementary: 4 hours for Cr/MWCNTs and1.5 hours for Fe/MWCNTs.
Point 5: What is the perspective of using these composite?
Response 5: Information about the perspectives of using these composites and corresponding reference outlined in Introduction:
"...the MWCNTs surface modification through inorganic and organic coatings deposition is an actula problem in modern material science. The solution of this problem is the deposition of nanosized continuous coatings of metal compounds on the outer surface of MWCNT by pyrolysis, having good adhesion to various fillers. In particular in our previous study the positive effects of coating MWCNTs with titanium carbide nanoparticles on the evolution of the reinforcement structure in bulk aluminum matrix nanocomposites was described [3]." - lines 60-66;
"Currently, intensive investigations have been performed on heterostructures prepared by decorating the MWCNTs surface with iron nanoparticles in order to obtain a material with the controlled magnetic properties, which can find various applications, for example, for the fabrication of nanoelectronic devices, in magnetic resonance imaging, and magnetic data storage [12]." - lines 78-82;
"Of great interest are pyrolytic Cr coatings, since they are characterized by high microhardness, heat resistance, hydrophobicity, and chemical resistance to hydrochloric and sulfuric acids and to alkali melt. Catalysts based on chromium oxide have found application in the ethylene polymerization [15], hydrocarbons dehydrogenation [16], methanol selective oxidation [17] and as sensors for ethanol vapor [18]." - lines 86-90.
Moreover, the short rewiev of possible application of these materials added to the Conclusion - lines 613-644.
Reviewer 3 Report
In this manuscript, authors report chemical and structural analyses of Cr or Fe-deposited MWCNTs. Unfortunately, in my opinion, the manuscript is not suitable for publication at this stage and would require an extensive revision. Although there are many figures in the manuscript, it is not easy to capture the main findings of this work. It is not easy to catch the novelty and significance. In fact, as stated in the introduction, synthesis of CNTs based on catalytic MOCVD and investigation of Fe or Cr with MWCNTs have been widely studied. The structure of this manuscript is not well organized. There are numerous awkward expressions and grammatical errors, too.
Author Response
Point 1: In this manuscript, authors report chemical and structural analyses of Cr or Fe-deposited MWCNTs. Unfortunately, in my opinion, the manuscript is not suitable for publication at this stage and would require an extensive revision.Although there are many figures in the manuscript, it is not easy to capture the main findings of this work. It is not easy to catch the novelty and significance.
Response 1:We suppose this remark is not correct. The article content, objects under study and research methods fully complies the requirements for publications in the special issue “Synchrotron Radiation techniques for the Investigation of Nanomaterials”. In the article, the authors report on the development and application of NEXAFS XPS synchrotron methods for the study of the pyrolytic chromium and iron nanoscale coatings on the MWCNT surface. These methods make it possible to study nanoscale coatings of nanostructured systems without destroying samples and to obtain unique information about their atomic-molecular composition and internal (latent) structure.
Point 2: In fact, as stated in the introduction, synthesis of CNTs based on catalytic MOCVD and investigation of Fe or Cr with MWCNTs have been widely studied.
Response 2:This remark is not quite correct. The introduction points out the need for further research:
“…The preparation of a hybrid material based on MWCNTs coated with thin layers of pyrolytic Cr and Fe is a relevant problem in view of possible application properties of such hybrids. Despite the intensive investigation of Fe/MWCNT and Cr/MWCNT composites [16-18], to date there are many open questions associated with both the technology of the synthesis of heterocomposites and the diagnostics of their physicochemical properties…..”.
Point 4: The structure of this manuscript is not well organized. There are numerous awkward expressions and grammatical errors, too.
Response 4: Since the authors of the article are not native English speakers, mistakes in style and grammar are present. So we plan to solve this problems by contacting the MDPI’s English Editing Service.
Also, we have made a number of additions to all sections of the article in accordance with specific proposals and comments of other reviewers.
Reviewer 4 Report
Suggestions in attached file.

Author Response
Point 1: The authors should revised manuscript according to Guide for authors attached in journal (eg. Tables, figures captions, etc.). Also, all Latin phrases (via, i.e., e.g., in-situ, etc...) in scientific writing should be in italics and abbreviations should be explained while using for the first time. Moreover, lines 6, 30 and other have different
text formatting.
Response 1: Comments have been taken into account. Text formatting fixed. Abbreviations problems solved.
Point 2: The authors should prepare Graphical Abstract to make the manuscript more attractive for potential readers, e.g. a scheme of MWCNTs coated with Cr and Fe preparation. The abstract is written in a very general way, the details about obtained materials should be mentioned.
Response 2: Graphical Abstract added.
Point 3: Examples of CNTs potential and well established applications should be presented. Also, already presented in literature modifications of MWCNTs, like Fe3O4/Ag should be presented.
Response 3: Examples of CNTs potential and well established applications added in Introduction:
"Composite based on MWCNTs decorated with Fe3O4 and polyethyleneimine-silver (nanocomposite Fe3O4-MWCNTs@PEI-Ag) was used as an efficient catalyst for chemoselective reduction of nitroaromatic and nitrile compounds [13]. Similar structures with Fe3O4 and silver nanoparticles on the of MWCNTs surface were used during the removal of a toxic model dye, methylene blue (MB), and an aromatic nitro compound, 4-nitrophenol (4-NP) [14]." - lines 82-86;
in section 3.5:
"A similar features of the nanotube coating by iron atoms is observed particularly for nanocomposite Fe3O4-MWCNTs@PEI-Ag [13]." - lines 471-472.
Point 4: Lines 528 and 544, repetitions.
Response 4: Fixed.
Point 5: More detailed characteristic of SEM and TEM/HRTEM images need to prepare.
The SEM and TEM/HRTEM methods were used only for composites testing. We think that there is no need to describe them in details. However, the additional information about the methods was added:
"The surface morphology of the synthesized materials was examined by SEM using Supra 50VP (Carl Zeiss AG, Germany) scanning electron microscope The structure of the MWCNTs, pyrolytic chromium and pyrolytic iron coatings grown by deposition from COL and Fe(CO)5 on the surface of the MWCNTs was studied by TEM with a LIBRA 200 MC Schottky Field emission gun instrument (Carl Zeiss AG, Germany) operating at 200 kV and the information resolution limit of 0.12 nm." - lines 189-194.
Point 6: Line 484: “Due to the hybridization of C1s and 3d orbitals of carbon and metal atoms, such a structure...” What does the author mean by hybridization of C1s and 3d orbitals? The authors should rephrase the statement so that readers can better understand the point the authors are trying to make.
Response 6: Orbital hybridization - the mixing of orbitals into new hybrid orbitals. It is a chemical term. New sentence: “Due to the mixing of C1s and 3d orbitals (hybridization) of carbon and metal atoms, such a structure...”
Point 7: Practically no discussion is present. The authors should discuss the advantages and disadvantages of their approach with other fabrication techniques and similar materials presented in the literature.
Response 7: We understand that there is no clear discussion in the manuscript, but there are discussion elements in the every subsections of section 3. According to this fact, we changed the names of these subsections.
Round 2
Reviewer 3 Report
Authors made reasonable changes.